# Preliminary Risk Assessment of Geological Disasters in Qinglong Gorge Scenic Area of Taihang Mountain with GIS Based on Analytic Hierarchy Process and Logistic Regression Model

**Ruixia Ma [1], Yan Lyu [1,*], Tianbao Chen [2] and Qian Zhang [1]**

[1] School of Geology Engineering and Geomatics, Chang'an University, Xi'an 710000, China;
2020126143@chd.edu.cn (R.M.); 2022226145@chd.edu.cn (Q.Z.)

[2] Chongqing Shutong Geotechnical Engineering Co., Ltd., Chongqing 401147, China; 13056507753@163.com

[*] Correspondence: lyuyan1118@163.com

**Abstract:** Qinglong Gorge Scenic Area (QGSA) boasts stunning natural landscapes, characterized by towering peaks and extensive cliffs. Nevertheless, the intricate geological backdrop and distinctive topographical conditions of this area give rise to various geological disasters, posing a substantial safety concern for tourists and presenting ongoing operational and safety management challenges for the scenic area. In light of these challenges, this study placed its focus on the geological disasters within QGSA and sought to assess risks across various scales. The assessment was accomplished through a combination of methods, including field surveys conducted in 2022, remote sensing interpretation, and comprehensive data collection and organization. For the geological disaster risk assessment of the scenic area, this research selected seven key indicators, encompassing terrain factors, geological elements, structural characteristics, and other relevant factors. The assessment utilized a logistic regression model, which yielded satisfactory results with an AUC value of 0.8338. Furthermore, a model was constructed incorporating seven indicators, encompassing factors such as population vulnerability, material susceptibility, and the vulnerability of tourism resources. To assess vulnerability to geological disasters, the Analytic Hierarchy Process (AHP) was employed, resulting in a CR of 0, thus ensuring the reliability of the findings. The outcomes of the risk assessment indicate that the low-risk area covers a substantial expanse of 5.45 km$^2$, representing 53.66% of the total area. The moderate-risk area extends over 3.59 km$^2$, constituting 35.43%, while the high-risk area encompasses 0.72 km$^2$, accounting for 7.14%. Additionally, the very high-risk area encompasses 0.38 km$^2$, making up 3.77% of the total area. Consequently, building upon the findings of the risk assessment, this paper introduces a risk classification and control prevention system. This system provides invaluable insights for disaster prevention and control in mountainous and canyon-type scenic areas.

**Keywords:** geological disaster; risk assessment; logistic model; analytic hierarchy process; Qinglong Gorge Scenic Area; Taihang Mountain; China

## 1. Introduction

Numerous renowned scenic areas, celebrated for their steep, rugged, treacherous, and extraordinary characteristics, are the products of natural geological processes. However, beneath their allure, they often conceal latent geological threats, including landslides, rockfalls, and mudslides. Extensive research underscores that geological disasters within scenic areas constitute an inherent and persistent reality, resistant to elimination or eradication. These associated risks endure over extended periods, necessitating the implementation of rational, effective, and practicable preventive measures. The exploration of geological

disaster risk assessment in scenic areas, as a significant non-engineering approach to mitigate disaster-related harm, stands as a strategic linchpin of proactive disaster prevention and reduction. It serves as the cornerstone for the advancement of scenic regions, the advancement of environmental preservation within the area, the formulation of emergency protocols, and the design of strategies for the prevention and control of geological disasters.

Scholars have dedicated their efforts to mitigating the impact of geological disasters on scenic areas through extensive research on the assessment of geological disaster risks. They have introduced the concept of geological disaster risk and risk assessment, emphasizing that the two primary aspects of risk assessment encompass the potential damage from and the consequences of disasters. Geological disaster risk is defined as the expected loss of life, property, and socio-economic stability resulting from a natural disaster within a specific area and timeframe. In essence, it can be expressed as "risk = hazard $\times$ vulnerability". Geological disaster risk assessment is founded on the concept of geological disaster risk and aims to quantitatively analyze and evaluate the likelihood and repercussions of geological disasters in areas at risk. It involves interdisciplinary approaches that equally consider the natural and social dimensions. These approaches primarily involve logistic regression analysis, fuzzy theory, extensibility theory, and gray system theory [1–7]. Transitioning from qualitative assessments, Blaikie [8] proposed a quantitative approach to disaster risk assessment, indicating that the sum of hazard and vulnerability calculations yields the risk assessment outcome. Shook [9] further emphasized that the risk assessment result is derived from the product of hazard and vulnerability. Scholars like Zhang, Xiang, and Ma underscored the importance of considering both the likelihood of disaster occurrence and the extent of damage in geological disaster risk assessment [10–12]. However, the inherent uncertainties in various facets of geological disasters render assessments, from spatial characteristics to disaster progression and consequences, highly uncertain [13]. With the continuous progress in risk management research, efforts in geological disaster prevention and mitigation have been further advanced [14–16]. Scholars such as Wu [17] and Zhang [18] conducted in-depth research on the theory and process of geological disaster risk assessment, leading to a substantial reduction in casualties resulting from geological disasters. In one instance, He [19] scrutinized the extent of danger posed by precarious rock formations, potential degrees of harm, the resilience of exposed structures, and the scope of devastation within the Jiuzhaigou scenic area. Risk assessment was quantified using the event tree method, culminating in the establishment of a model for assessing the risk associated with hazardous rock formations. Similarly, Han [20] computed the risk of geological disasters by constructing a model based on the cumulative function encompassing vulnerability, exposure, and hazard. In the context of scenic areas built upon tourism resources, the evaluation of rockfall disaster risks must comprehensively encompass the potential harm to these resources. Consequently, numerous scholars [21–26] have undertaken risk assessments with a specific focus on tourism safety. For example, Sun [27] introduced a risk assessment model that includes elements such as "hazard, exposure, vulnerability, and disaster prevention and mitigation capacity" relevant to disasters involving tourism resources.

Geographic Information Systems (GIS), initially introduced by Garrison, have found extensive application in the field of geological disasters. Geological disasters are the consequence of the interplay between disaster-prone environments and triggering factors, and they exhibit a strong correlation with spatial information. GIS technology serves as a valuable tool for the management of diverse geological disasters and their associated data. It enables the exploration of statistical relationships between the occurrence of geological disasters and environmental factors, both in spatial and temporal dimensions. This facilitates the assessment of the likelihood of various geological disasters and their potential repercussions. Furthermore, GIS empowers the storage of geological disaster data within geographic databases, facilitating spatial analysis and the creation of two-dimensional and three-dimensional visualizations. Anbalagan [28] introduced an innovative approach to risk assessment by developing a risk assessment matrix, drawing insights from the study of

landslide disasters. Pei [29], Ruan [30], and Ni [31] combined GIS with Certainty Factor (CF) and Analytic Hierarchy Process (AHP) to undertake geological disaster risk assessment. Zhu [32] established a geological disaster risk assessment system grounded in GIS and conducted a nationwide risk assessment for landslide disasters. Sun [33] conducted safety risk assessment utilizing GIS technology, employing tools such as fishbone diagrams and dynamic Bayesian methods for the Changbai Mountain Scenic Area. Meanwhile, Luo [34] performed hazard and vulnerability assessments of the Jiuzhaigou Scenic Area, employing a GIS platform in conjunction with CF and AHP.

QGSA is situated in the eastern part of the Taihang Mountains within Huguan County, Shanxi Province. This region abounds in tourism resources, boasts well-developed community service facilities, and supports a high population density. It holds a prominent position as one of the key areas within the Taihangshan Grand Canyon National Geopark. However, due to its proximity to the frontal fault of the Taihang Mountains, the region experiences intense tectonic activity and severe surface erosion. Geological disasters occur frequently under the influence of the unique mountain climate, posing a significant threat to local residents, tourists, and the area's tourism infrastructure. Therefore, this study, informed by comprehensive field surveys, conducts a meticulous analysis of the spatial distribution patterns of geological disasters and the factors that influence them. It formulates a model for assessing geological disaster risk and vulnerability using a combination of logistic regression and the Analytic Hierarchy Process (AHP). Additionally, the research carries out a comprehensive geological disaster risk assessment for QGSA on a GIS platform, categorizing the levels of geological disaster risk within the region. This research addresses a significant gap in the current scenic area of geological disaster risk studies in scenic areas in China, while simultaneously addressing the pressing need for disaster prevention and mitigation in such picturesque locales.

## 2. Geological Environmental Overview and Data Source

### 2.1. Physical Geography and Geological Conditions

QGSA of Taihang Mountain is located in the transitional zone extending from the southern foothills of Taihang Mountain to Linzhou Basin. It is renowned for its water features, canyon topography, and rock strata profiles. The scenic area exhibits notable variations in elevation and experiences significant climatic diversity. Moreover, the canyons within this area possess distinctive microclimatic characteristics, with an average annual atmospheric precipitation of 543.8 mm, primarily concentrated between June and September. The drainage area of QGSA is primarily composed of the east-west valley formed by the Suburban River and the north-northwest valleys, with the two valleys forming a "Y"-shaped layout in the plan view, creating a dendritic water system (Figure 1). The topography is characterized by steep mountains and deep canyons, featuring a pronounced west-to-east slope, with elevation variations spanning from 523 m to 1303 m and a relative elevation difference of up to 780 m. The average slope exceeds 70°. The exposed geological strata consist mainly of sedimentary cover layers and Precambrian intrusive bodies. The Precambrian intrusive bodies are chiefly composed of dark cloud (hornblende) diorite intrusions, while the sedimentary cover layers encompass the Middle Proterozoic Great Wall Formation, the Lower Paleozoic Cambrian Formation and Ordovician Formation, in addition to the Quaternary Fourth Formation.

The study area is situated in the southeastern sector of the Taihang Block, encompassed within the Lvliang-Taihang Fault Block. It lies in close proximity to the Linzhou Fault Depression Basin to the east and is predominantly governed by a vast and intricate anticline structure. This region is notably characterized by the presence of 16 nearly parallel faults, with the majority manifesting as normal faults. These fault lines primarily align in a north-northeastern direction. Among these, the principal controlling fault is the Huangyadi-Yangjiachi-Bada Normal Fault (Figure 1). The area experiences intricate tectonic activities marked by extensive faulting. The influence of tectonic stress induces the development

of joint fractures within rock masses, creating conditions conducive to the occurrence of geological disasters.

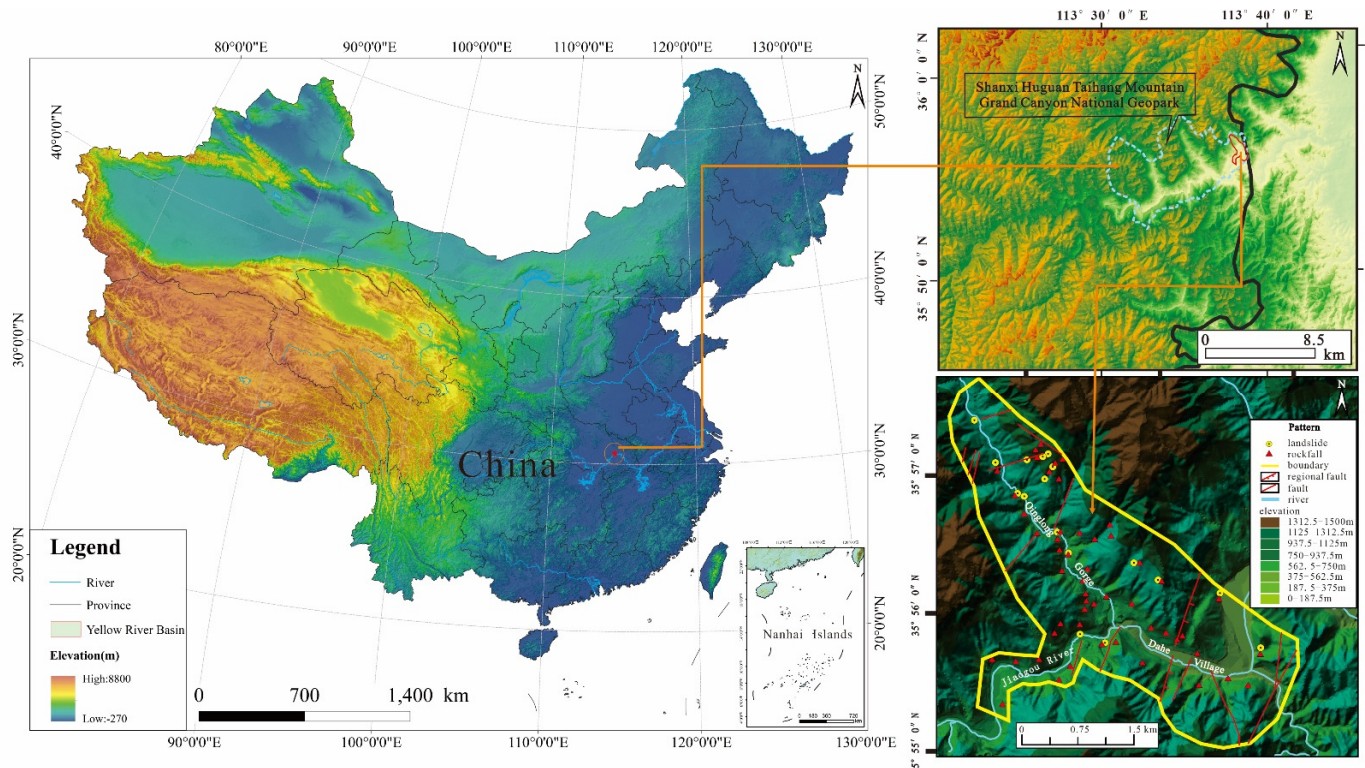

**Figure 1.** Location map of study area.

### 2.2. Data Sources

By utilizing drone photographs, Google Earth remote sensing imagery, and high-precision remote sensing data with a remarkable 0.2-m resolution, we incorporated on-site slope-by-slope inspections to delineate the study area (Figure 2). Employing 1:50,000 geological maps and geological disaster distribution charts, we executed a comprehensive workflow encompassing interpretation, validation, re-interpretation, and re-validation processes. This rigorous methodology unveiled a total of 69 geological disasters within the region, with a predominant concentration within the valleys. Among these, 17 were identified as landslides, constituting 24.64%, while 52 were categorized as collapses, representing 75.36% (Figure 1).

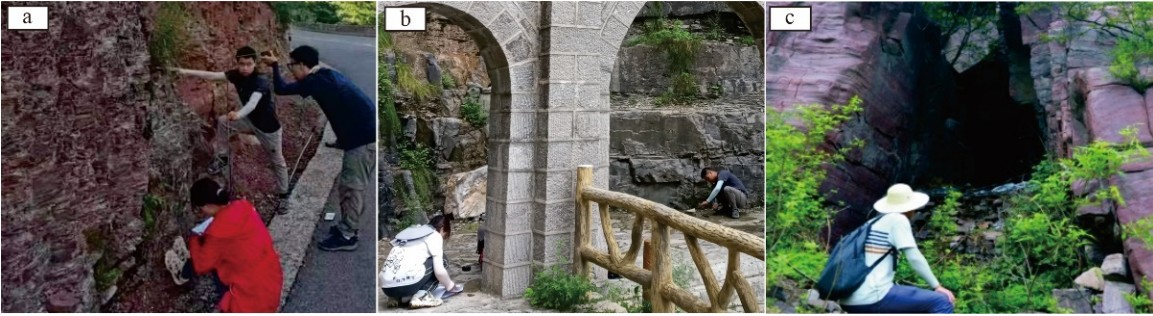

**Figure 2.** The figure of field work. (**a**) Structural plane measurement; (**b**) on-site rockfall statistics; (**c**) field survey.

Through the integration of spaceborne remote sensing imagery, low-altitude sensing photogrammetry, and ground-based LiDAR detection, geological disaster data were acquired from multiple perspectives, encompassing the sky, space, and ground. This amalgamation culminated in the development of a comprehensive and high-precision geological disaster information model, facilitating efficient and dependable identification of geological disasters. Simultaneously, this methodology surmounted the constraints imposed by terrain and topography, effectively achieving comprehensive ground geological disaster detection.

## 3. Development Characteristics and Influencing Factors of Geological Disasters

### 3.1. Development Characteristics of Geological Disasters in QGSA

Using remote sensing interpretation, three-dimensional modeling, and comprehensive on-site inspections, we systematically classified and statistically analyzed the 69 geological disasters within the scenic area based on their elevation and scale. The findings indicated that collapses mainly occur at elevations greater than 80 m and slopes steeper than 60 degrees (Figure 3a). The majority are of small scale, characterized by rockfalls with relatively small volumes, and are distributed along valleys and roads. Medium- and large-scale collapses each account for less than 10% of the total and are mainly located near fault zones. Different rock types exhibit significant differences in the size and quantity of collapsed rocks, with quarzitic sandstone and limestone formations having larger and fewer rock fragments, while shale and mudstone formations have smaller and looser rock fragments (Figure 3b). Landslides primarily occur on steep slopes with gradients ranging from 25° to 80°. Small-scale landslides predominate, followed by medium-scale ones, and are composed mainly of loose Quaternary debris and alluvium (Figure 3c). Additionally, a gigantic conformable rockslide has developed within the scenic area, spanning multiple stratigraphic layers such as the Zhangxia Formation, Gushan Formation, and Mantou Formation. It was formed under the dual control of faulting and karst processes.

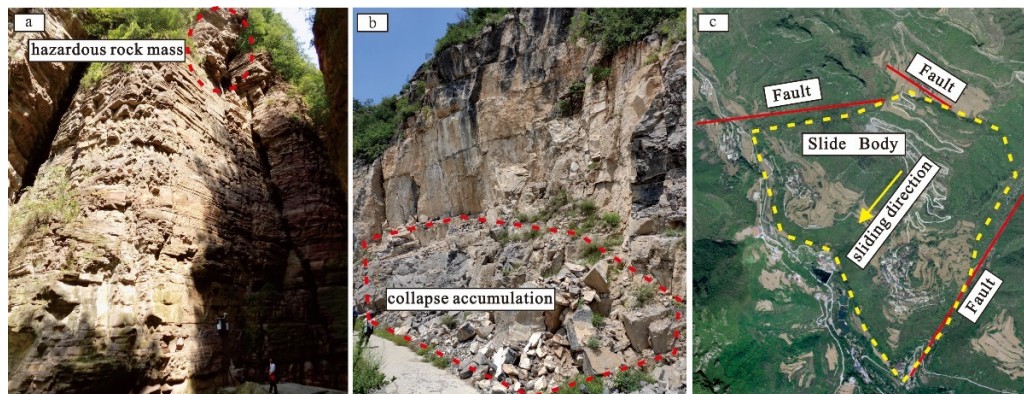

**Figure 3.** Typical geological disasters and hidden dangers in QGSA. (**a**) Hazardous rock mass; (**b**) collapse accumulation; (**c**) slide body.

### 3.2. Distribution Patterns of Geological Disasters in QGSA

The spatial distribution of geological disasters is intimately linked to river incision, canyon evolution, and human engineering activities. Within QGSA, these geological disasters are notably concentrated on both sides of the scenic roads and along the Suburban River valley (Figure 1). Qinglong Gorge showcases river valleys formed during various periods, serving as a primary landscape feature within the scenic area while also creating conducive conditions for the development of geological disasters. Subsequently, numerous small-scale geological disasters have emerged on both sides. Furthermore, the construction of roads within the scenic area has perturbed the rock and soil, leading to the linear distribution of geological disasters along the scenic roads, which exhibit a distinctive branching pattern.

### 3.3. The Return Period for Geological Disasters in QGSA

Geological disasters in QGSA exhibit a relatively concentrated formation period. The temporal distribution of geological disasters in QGSA is chiefly influenced by rainfall and freeze–thaw processes. High-frequency and clustered geological disasters tend to transpire during periods of heavy summer rains and spring freeze–thaw events (Figure 4). Within the scenic area, summer precipitation is notably concentrated, while the spring freeze–thaw phase prompts the melting of snow and ice, leading to water infiltration into the mudstone. This infiltration, in turn, causes the mudstone to soften and liquefy, significantly diminishing the shear strength of the rock mass. Consequently, the rock mass undergoes structural surface displacement along weak planes, rendering it vulnerable to geological disasters.

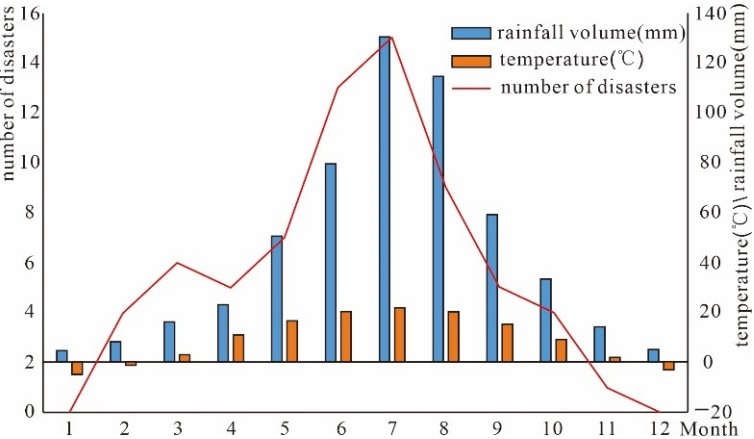

**Figure 4.** Law of disaster time distribution.

### 3.4. Influencing Factors of Geological Disasters in QGSA

3.4.1. Elevation

Elevation plays a pivotal role in determining the magnitude of stress experienced by rock masses. Stress levels escalate with the rising elevation of slopes, thereby impacting the potential energy associated with collapses and landslides. Figure 5 illustrates that geological disasters in QGSA are predominantly concentrated within the elevation range of 523 m to 816 m, encompassing an area that constitutes 58.95% of the total scenic area and encompasses 76.82% of all geological disasters in the region. In contrast, the occurrence of geological disasters significantly diminishes in other elevation brackets. The elevation ranges of 816 m to 963 m and 963 m to 1303 m correspond to 30.47% and 10.58% of the total scenic area, respectively, with geological disasters in each bracket accounting for 11.59%.

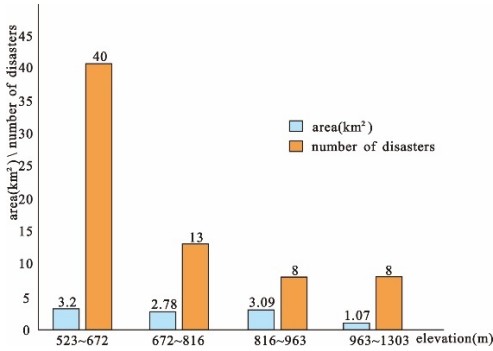

**Figure 5.** Statistics of elevation and disasters.

### 3.4.2. Slope Gradient

Slope gradient is a critical factor in assessing the risk of geological disasters. Gentle slopes, with their low shear stress, exhibit a reduced likelihood of experiencing such disasters. Conversely, an increase in slope gradient elevates stress levels, substantially augmenting the probability of geological disasters. Utilizing DEM data for QGSA, the region was categorized into five segments based on slope gradient. As shown in Figure 6, the area with slopes less than 65° within Qinglong Gore accounted for 15.78% of the entire scenic area, and the proportion of geological disasters occurring in this region was 20.29%. In contrast, the area with slope gradients greater than 65°, characterized by vertical rock walls, covered a high percentage of 84.22% of the total area, with 79.71% of geological disasters occurring in this region.

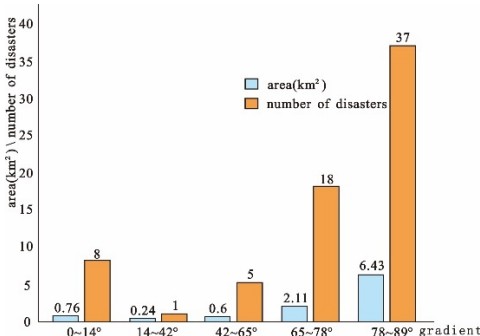

**Figure 6.** Statistics of the scope of slope and disasters.

### 3.4.3. Stratum Lithologic

The exposed geological formations in QGSA primarily consist of the following rock types: dark cloud (hornblende) diorite intrusive rocks (Arz), quarzitic sandstone (Chd), thin-layered mudstone and shale (Chz), and massive layered quarzitic sandstone (Chc). There are also mudstone (Єm) and shale (Єm), as well as various types of limestone (Єz, Єg, Om), dolomite (Єs-Os), and Quaternary alluvial and slope deposits (Figure 7). The regional strata exhibit distinct alternations between soft and hard rocks, providing the material foundation for the occurrence of geological disasters.

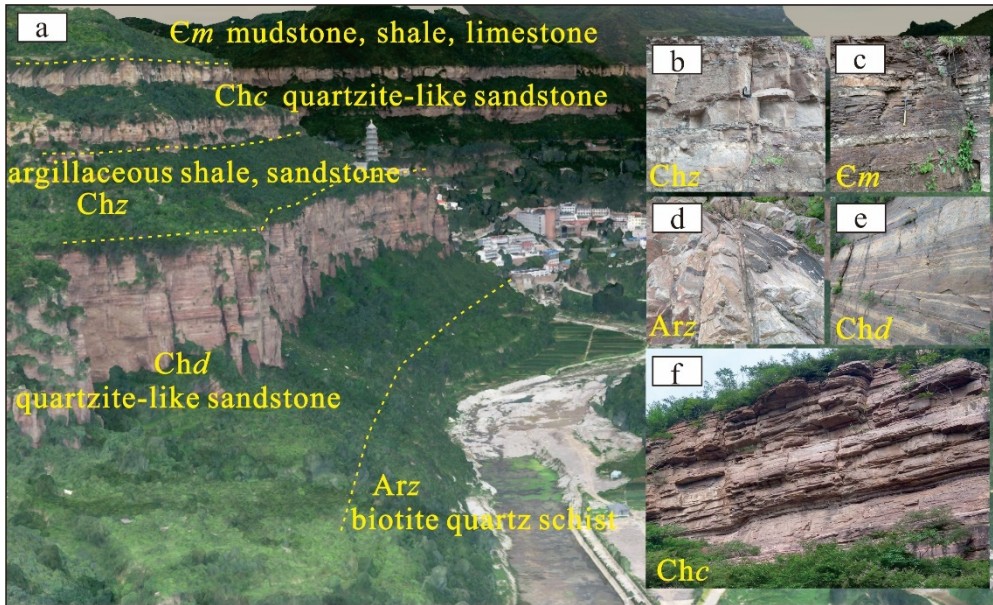

**Figure 7.** Map of stylus rock distributed in QGSA. (**a**) Stratigraphic panorama; (**b**) the stratigraphic of Chz; (**c**) the stratigraphic of Єm (**d**) the stratigraphic of Arz; (**e**) the stratigraphic of Chd; (**f**) the stratigraphic of Chc.

### 3.4.4. Distance from Fault

QGSA features the development of two relatively large fault zones, namely the north-northeast-trending and northwest-trending fault zones (Figure 1). Based on survey statistics, it has been observed that the orientation of joints within the area is consistent with the direction of these fault zones, indicating that the development of rock joints is primarily controlled by these two fault zones. The ongoing activity of these fault zones can exacerbate the opening of rock joints and, at the same time, enhance the erosion of weak interlayers within the geological strata by groundwater in the slopes, which can lead to the occurrence of geological disasters.

### 3.4.5. Rainfall

QGSA has a complex topography and diverse landforms, with significant differences in elevation and clear characteristics of a mountainous canyon microclimate. Precipitation is concentrated in the summer and autumn seasons (Figure 3), with an average annual rainfall of 543.8 mm. Rainfall exhibits significant interannual variability, with the highest recorded rainfall reaching 908 mm and the lowest reaching 335.6 mm. Abundant rainfall can significantly reduce the frictional resistance between rock layers, thereby accelerating the movement of landslides and potentially hazardous rock masses, leading to the occurrence of geological disasters.

### 3.4.6. Human Engineering Activity

The development of tourism resources, bridge construction, road building, and mining activities in QGSA has caused certain disturbances to the mountain slope structures. In QGSA, activities such as the construction of tourist trails, parking lots, visitor centers, concentrated residential areas in Dahe Village, construction of houses and roads at the foothills, and other activities are frequent. As a result of these engineering disturbances, some rock and soil masses have experienced changes in slope stress conditions, making them potential geological disaster-prone areas.

## 4. Geological Disaster Risk Assessment of QGSA

### 4.1. Hazard Assessment of Geological Disasters

4.1.1. Selection and Classification of Hazard Assessment Factors

The occurrence of geological disasters is the result of the combined effects of internal controlling factors within the slope and triggering factors from the external environment. Through the analysis of the development characteristics of geological disasters in QGSA and the environmental factors conducive to disasters, a set of assessment factors, including disaster density, elevation, slope, lithology, faults, rainfall, and human engineering activities, were selected to construct the index system for the assessment of geological disaster hazards in QGSA. Considering the present geological disaster conditions in the study area, this paper partitions the DEM grid of QGSA into 4715 columns and 4992 rows, resulting in a total of 10,145,365 units. Qualitative parameters, including rainfall, lithology, faults, and elevation, are directly normalized to a range of [0, 1]. As for factors like disaster density, slope, and human engineering activities, the fuzzy membership tool in ArcGIS is used for normalization. The classification value type selected is a linear function, and the computations adhere to Equation (1), as illustrated in Figure 8.

$$y_i = \frac{x_i - x_{min}}{x_{max} - x_{min}} \tag{1}$$

(1)    Disaster density

Utilizing the distribution map of disasters derived from field surveys, the disaster point density is categorized into four density intervals using the natural break method, with the following size sequences: 0.0–0.0062, 0.0062–0.0159, and 0.0159–0.0286 (Figure 8a).

(2) Elevation

Elevation reflects topographic features, and areas at varying elevations manifest distinctions in rainfall, temperature, and geological traits. Within the ArcGIS platform, the study area's elevation is categorized into four intervals utilizing the natural break method: 523–672 m, 672–816 m, 816–963 m, and 963–1303 m (Figure 8b).

(3) Slope

A steeper slope indicates reduced slope stability due to a greater concentration of shear stress at the slope's base, leading to an increased susceptibility to geological hazards. Moreover, slope steepness impacts surface runoff, slope seepage drainage, and vegetation growth, which, in turn, influence slope stability. Within ArcGIS, the reclassification tool is applied to segment the slope raster data into five slope intervals, using the natural break method: 0–14°, 14–42°, 42–65°, 65–78°, and 78–89° (Figure 8c).

(4) Lithology

Stratigraphic lithology not only governs the material composition of geological disasters but also exerts a significant influence on the deformation, failure characteristics, and strength of these disaster bodies. An evaluation of the stratigraphic lithology in the scenic area enables the categorization of strata hazard levels into five tiers: strong, relatively strong, moderate, general, and weak (Figure 8d).

(5) Faults

Expert research indicates that closer proximity to a fault corresponds to intensified tectonic activity and an increased occurrence of geological disasters. Consequently, this study assesses the potential for geological disasters in the area by considering its distance from the fault. Utilizing ArcGIS 10.2 software, Euclidean distance analysis and the subsequent reclassification of the scenic area divides it into four regions: <100 m, 100–500 m, 500–1000 m, and >1000 m (Figure 8e).

(6) Rainfall

Increased heavy rainfall frequency is associated with a higher incidence of geological disasters like collapses and landslides. This paper divides the study area into four regions based on the annual rainfall contour lines from the Huguan County rainfall contour map: <650 mm, 650–648 mm, 648–646 mm, and >646 mm (Figure 8f).

(7) Human engineering activities

This paper classifies human engineering activities into three categories: road construction, residential housing construction, and tourist development construction. ArcGIS is employed to calculate and produce a map illustrating the correlation between human engineering activities and the distribution of geological disaster points (Figure 8g).

### 4.1.2. The Determination of Logistic Regression Model and Assessment Factor Weight

The multiple linear regression method requires that the dependent variable must be continuous and there must be a clear linear trend with the independent variables. Since the occurrence of geological disasters belongs to a Boolean relationship, it cannot meet the requirements of multiple linear regression. Therefore, it is necessary to perform a logistic transformation on the original multiple linear regression function. The form of the logistic regression model function is as follows:

$$P = 1/(1 + e^{(-g(x))}) \tag{2}$$

where $g(x) = \beta_0 + \beta_1 x_1 + \beta_2 x_2 + \ldots + \beta_i x_i$ ($i = 0, 1, 2, \ldots, n$), P represents the probability of geological disaster occurrence, and i represents the number of evaluation factors. Therefore, the odds ratio of geological disaster occurrence to non-occurrence is:

$$(P(y = 1 \mid X))/(P(y = 0 \mid X)) = p/(1 - p) = e^{(g(x))} \qquad (3)$$

Taking the logarithm on both sides of Equation (3), we obtain $\ln(p/(1 - p)) = g(x) = \beta_0 + \beta_1 x_1 + \beta_2 x_2 + \ldots + \beta_i x_i$. In the assessment of geological hazard risk, the factor weight values are determined by $\beta_i$ in function $g(x)$.

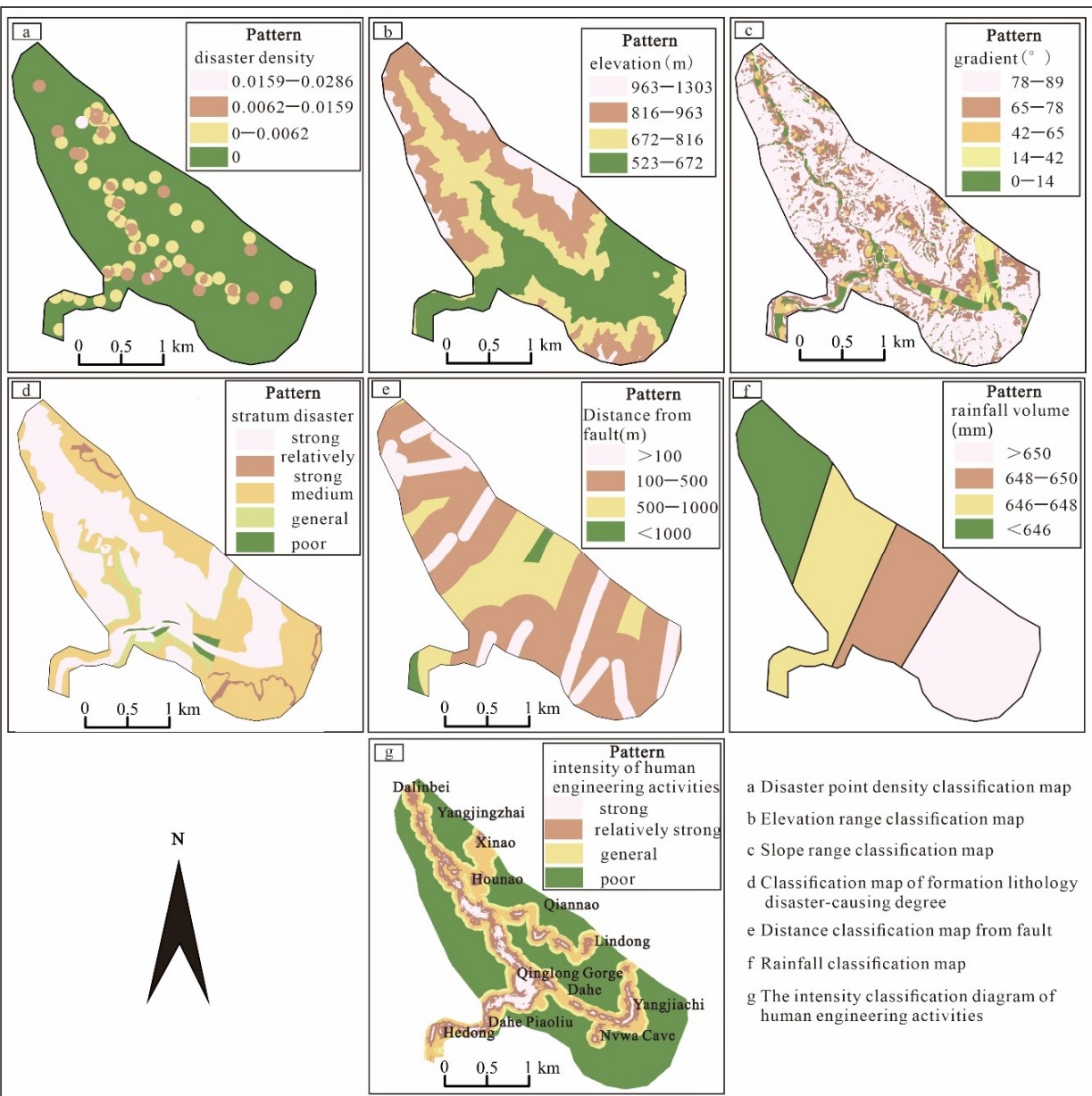

**Figure 8.** Classification map of geological disaster hazard assessment factors.

Using whether geological disasters occur as the dependent variable (0 indicating no geological disaster occurrence and 1 indicating geological disaster occurrence), with evaluation factors as independent variables, geological disasters are subjected to logistic regression analysis using SPSS software 26. The weights of the evaluation factors are obtained (Table 1), listed in descending order as follows: faults (0.1780), geological formation (0.1712), human engineering activities (0.1652), hazard density (0.1510), elevation (0.1313), slope (0.1051), and precipitation (0.0982).

**Table 1.** Geological disaster geohazard assessment factors' weight.

| Evaluation Factor | Stratum Lithologic | Distance from Fault | Elevation | Slope Gradient | Human Engineering Activity | Geohazard Density | Rainfall |
|---|---|---|---|---|---|---|---|
| Regression coefficient | 1.468 | 1.526 | −1.126 | −0.901 | 1.417 | 1.295 | −0.842 |
| Significance | 0.000 | 0.000 | 0.000 | 0.000 | 0.000 | 0.010 | 0.025 |
| Weight | 0.1712 | 0.1780 | 0.1313 | 0.1051 | 0.1652 | 0.1510 | 0.0982 |

### 4.1.3. Analysis and Test of Hazard Assessment Results

The geological disaster hazard assessment index for QGSA is computed based on the normalized values and their respective weights. In GIS, the results of these superimposed calculations are categorized into four hazard value intervals using the natural breaks method: 0.28–0.43, 0.43–0.49, 0.49–0.57, and 0.57–0.76. A higher hazard value indicates a greater level of hazard in the area. By considering the hazard value intervals and the actual development of geological disasters in the scenic area, QGSA is categorized into four zones using cluster analysis: extremely high-hazard value zone, high-hazard value zone, medium-hazard value zone, and low-hazard value zone (Figure 9a). The statistics of the geological disaster hazard assessment results for QGSA are presented in Figure 9b,c, indicating that the low-hazard value zone covers an area of 2.43 km$^2$, constituting 23.96% of the entire scenic area; the medium-hazard value zone spans 2.91 km$^2$, representing 28.71% of the entire scenic area; the high-hazard value zone and extremely high-hazard value zone encompass areas of 2.79 km$^2$ and 2.01 km$^2$, respectively, accounting for 27.51% and 19.82%.

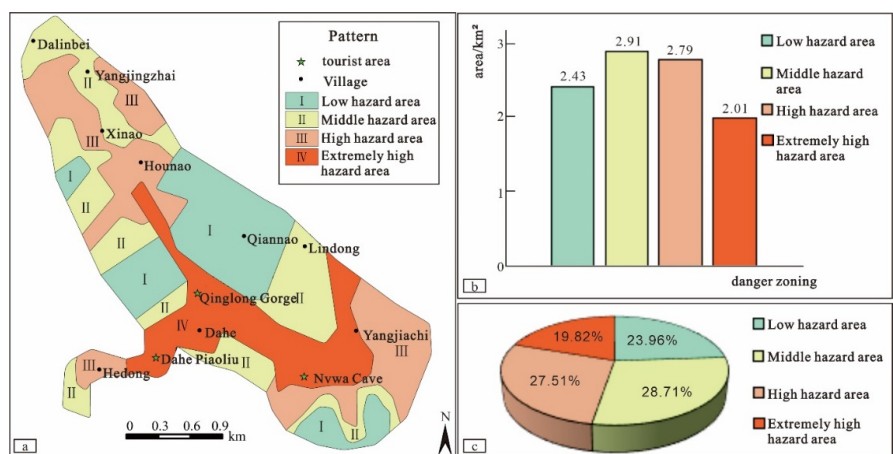

**Figure 9.** Hazard assessment zoning results in QGSA. (**a**) Hazard assessment zoning; (**b**) area chart; (**c**) area percentage chart.

The extremely high-hazard value zone exhibits the development of 31 geological disasters, representing a significant percentage of 44.93%. These disasters are predominantly concentrated in several key areas, including the Qinglong Gorge tourist area, Dahe Piaoliu, the Nuwa Cave tourist area, and the Hedong-Dahe Village-Yangjiachi Village region. This zone encompasses a diverse topography, featuring erosion-prone low mountain regions and deposited river valleys, which are conducive to human habitation. Frequent engineering construction activities have a pronounced impact on the stability of the rock masses in this area.

The high-hazard value zone presents 24 geological disasters, constituting 34.78% of the total. These disasters are primarily distributed in Hedong Village, the eastern part of Yangjiachi, and the northern section of QGSA. The overall elevation in this zone is relatively high, with Hedong Village situated in a deposited river valley area, while the other two regions are characterized by karst erosion and middle mountain terrain. These areas display

fractured rock masses with developed joints and rock cavities. Intensive road excavation activities significantly contribute to the induction of geological disasters in this zone.

The medium-hazard value zone features six geological disasters, accounting for 8.70%. These disasters are predominantly distributed in villages such as Dalinbei, Yangjingzhai, and Lingdong. The terrain in this zone is primarily composed of karst erosion and middle mountain areas. The level of human engineering activities is relatively low, resulting in minimal disturbance to rock stability.

The low-hazard value zone experiences the development of eight geological disasters, constituting 11.59% of the total. These disasters are mainly distributed on both sides of the Qinglong Gorge tourist area, including Xinao and Qiannao Village. The elevation in this area is relatively high, with the terrain dominated by karst erosion and middle mountain regions. In this zone, rock stability is notably high, and the intensity of human engineering activities is low, making it less susceptible to geological disasters.

The precision of geological disaster hazard assessment results significantly influences the reliability of the model. Therefore, it is imperative to scrutinize the accuracy of these assessment outcomes. Conforming to established guidelines, sample sizes less than 100 introduce a larger margin of error, while exceeding 500 is deemed satisfactory to meet accuracy requirements. In this research, the grid units within the study area were meticulously partitioned into 1 m × 1 m dimensions, comprising 4715 columns, 4992 rows, and a total of 10,145,365 grid units. Among these, a random sample of 10,000 grid-unit data points was selected as the sample size for the independent variables. Utilizing the collinearity diagnostic function in SPSS software 26, an assessment of the independence of the evaluation factors was conducted based on the calculated Variance Inflation Factor (VIF), ensuring that it would not impact the model results. Simultaneously, the ROC curve emerged as a widely adopted method for evaluating the reliability of geological disaster hazard assessment models (Figure 10). The AUC value, representing the area enclosed by the ROC curve and the *x*-axis, provides a critical measure. The closer its value is to 1, the higher the reliability of the hazard assessment model. The AUC value of the geological disaster hazard assessment model formulated in this paper for QGSA stands at 76.20%. This underscores the relatively high accuracy of the assessment model.

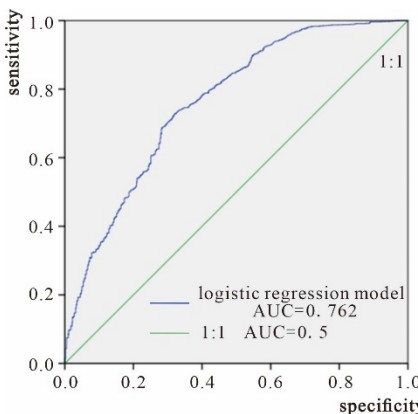

**Figure 10.** ROC curve of geological disaster risk evaluation in QGSA.

### 4.2. Vulnerability Analysis of Geological Disasters

4.2.1. Selection and Classification of Vulnerability Evaluation Factors

Taking into full consideration the characteristics of the geological disaster-bearing bodies in QGSA, this paper selects seven evaluation factors, including vulnerability of resident population, vulnerability of tourist population, vulnerability of residential buildings, vulnerability of roads and bridges, vulnerability of tourist facilities, vulnerability of cultural landscapes, and vulnerability of natural landscapes, to construct the vulnerability assessment index system for geological disasters. Based on actual survey results and remote sensing images, each evaluation factor is graded (Figure 11). Among them, QGSA

is divided into 20 residential areas according to villages, with a maximum population density of 1.58 people/100 m$^2$ and a minimum of 0.06 people/100 m$^2$ (Figure 11a). Tourist populations consist of tourists within scenic spots and those in hotels and folk customs, counted uniformly using a people flow counter, with a maximum flow of 300 people/hour (Figure 11b). There are a total of 87 residential building areas in QGSA, with buildings of three structural types: reinforced concrete, steel and concrete, and brick and wood. Residential buildings are classified into six levels based on different structures (Figure 11c). The transportation network of Qinglong Gorge consists of the S327 provincial road and county, township, and village roads. The vulnerability value of roads is determined by the unit-length road value. Therefore, the road value within the scenic area is divided into three levels according to relevant standards (Figure 11d). To ensure visitors' touring experience, the scenic area is equipped with a large number of tourist facilities, including pavilions, parking lots, geological museums, Buddha stupas, tourist trails, etc. Hence, the value of tourist facilities within the scenic area is divided into seven levels according to relevant standards (Figure 11e). QGSA has a total of 20 natural landscapes and 11 cultural landscapes, which are the mainstay of the scenic area and play a significant role in vulnerability assessment. Therefore, this paper selects this factor and classifies it based on landscape density, with cultural landscapes and natural landscapes classified into four levels (Figure 11f) and six levels (Figure 11g).

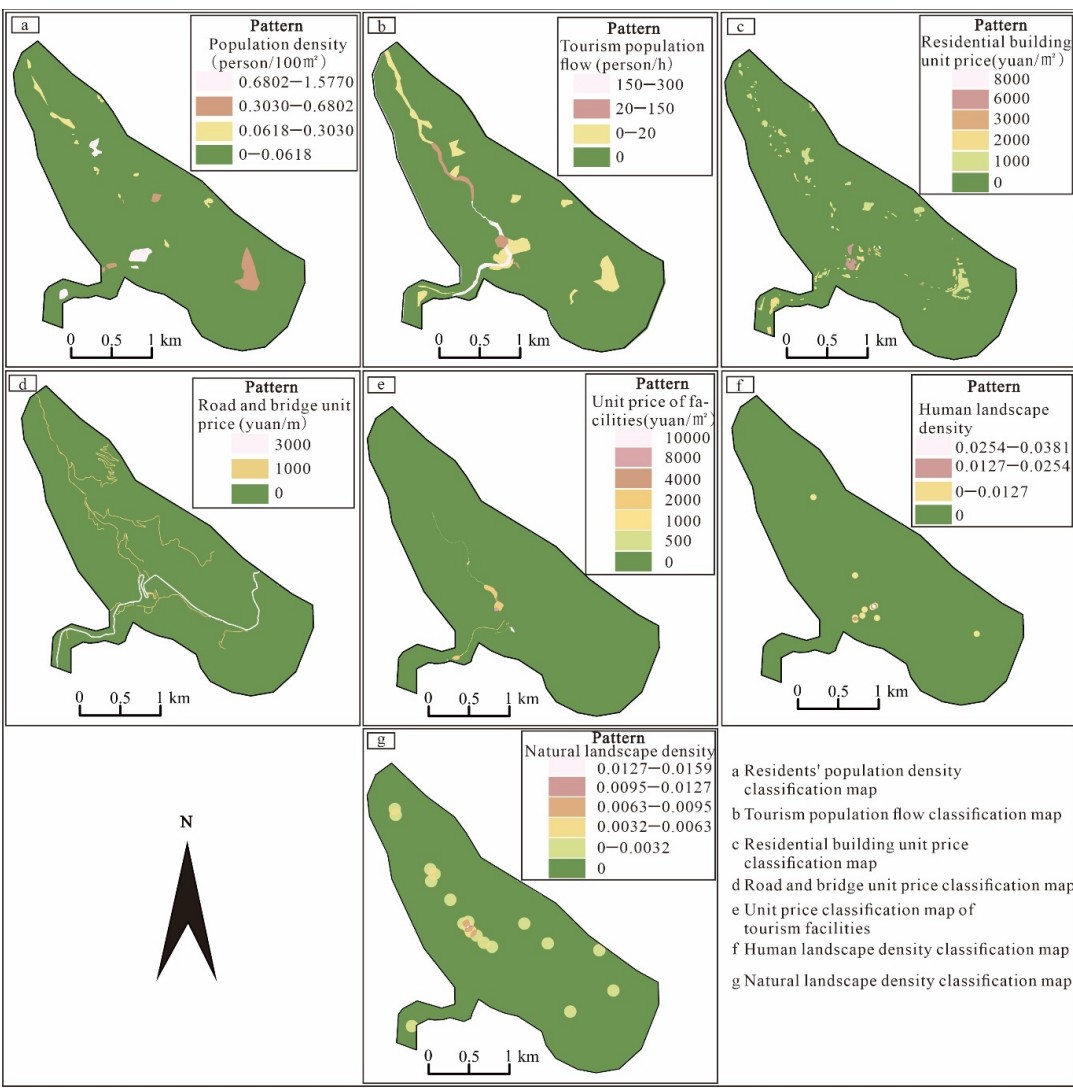

**Figure 11.** Classification map of vulnerability assessment factors.

### 4.2.2. Determination of AHP and Evaluation Factor Weight

With the goal of assessing the vulnerability of geological disasters in QGSA, population vulnerability, material vulnerability, and tourism resource vulnerability are taken as criteria layers, and the above-mentioned seven evaluation factors are taken as scheme layers to construct the evaluation model. The scaling values for factors based on importance and membership are determined through expert scoring to create a judgment matrix. According to yaahp software 2.9, the maximum eigen root $\lambda_{max}$ of the matrix is obtained, the CI value is obtained according to Equations (4) and (5), and the inconsistency of the factors in the matrix is judged to be within the allowable range. The weight value of the assessment factor is calculated according to Equation (6) (Table 2). In descending order, the weights are as follows: tourist population (0.5833), resident population (0.1944), natural landscapes (0.0833), roads and bridges (0.060), tourist facilities (0.0330), cultural landscapes (0.0278), and residential buildings (0.0182).

$$CI = (\lambda - n)/(n - 1) \tag{4}$$

$$CR = CI/RI \tag{5}$$

$$W_i = \frac{1}{n}\sum_{j=1}^{n} \frac{a_{ij}}{\sum_{h=1}^{n} a_{h1}} \tag{6}$$

where $W_i$ is the weight value, $a_{h1}$ represents the element of the first column of row h, and $a_{ij}$ represents the element of column j of row i.

**Table 2.** The table of geological disaster vulnerability assessment factors' weight.

| Criterion Layer | Population Vulnerability | | Material Vulnerability | | | Vulnerability of Tourism Resources | |
|---|---|---|---|---|---|---|---|
| **Schematic Layer** | **Resident Population** | **Tourism Population** | **Tourist Facilities** | **Residential Structure** | **Road, Bridge** | **Cultural Landscape** | **Natural Landscape** |
| Scheme layer weight | 0.25 | 0.75 | 0.2970 | 0.1634 | 0.5936 | 0.25 | 0.75 |
| Consistency test of scheme layer weight | $C_R = 0$ | | $C_R = 0.0088$ | | | $C_R = 0$ | |
| Criterion layer weight | 0.7778 | | 0.1111 | | | 0.1111 | |
| Criterion layer weight consistency test | $C_R = 0$ | | | | | | |
| Index weight | 0.1944 | 0.5833 | 0.0330 | 0.0182 | 0.0600 | 0.0278 | 0.0833 |

### 4.2.3. Vulnerability Evaluation Results

Utilizing the normalized values and their corresponding weights, the vulnerability assessment index for geological disasters in QGSA is meticulously computed. In GIS, the results of overlay calculations are parsed into four vulnerability value intervals using the natural breaks method: 0–0.03, 0.03–0.12, 0.12–0.29, and 0.29–0.66. The ascending vulnerability values indicate a higher susceptibility of the area to geological disasters. By amalgamating the vulnerability value intervals with the actual QGSA distribution, cluster analysis is employed to classify the area into four regions: extremely high vulnerability, high vulnerability, moderate vulnerability, and low vulnerability (Figure 12a). The statistics for the vulnerability assessment of geological disasters in QGSA are detailed in Figure 12b,c. These results illustrate that:

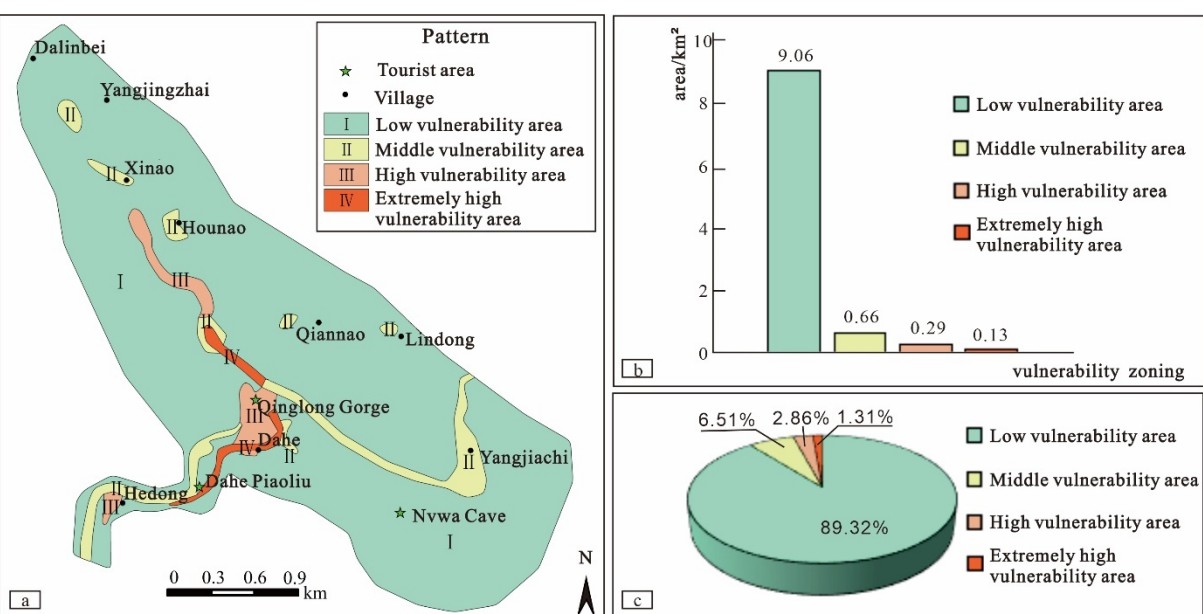

**Figure 12.** Vulnerability assessment zoning result in QGSA. (**a**) Vulnerability assessment zoning; (**b**) area chart; (**c**) area percentage chart.

The low vulnerability area covers 9.06 km$^2$, constituting 89.32% of the entire scenic area. Located in high mountain gorge areas, it is unsuitable for human habitation, with a limited number of small villages and a sparse population. Large-scale road construction is absent.

The moderate vulnerability area spans 0.66 km$^2$, encompassing 6.51% of the entire scenic area. This region includes villages with relatively low population density and areas on both sides of the S327 provincial road. The focus is on the vulnerability of the resident population, residential buildings, and road bridges. The population is scattered, and houses are constructed from brick and wood.

The high vulnerability area covers 0.29 km$^2$, accounting for 2.86%. It mainly includes Dahe, Hedong, and the area along Xiaojiuzhai-Qinglongpu-Qingrenhu. These villages have a substantial resident population, and the scenic area boasts abundant tourist resources with a high influx of tourists.

The extremely high vulnerability area spans 0.13 km$^2$, making up 1.31% of the total. It predominantly encompasses Yixiantian in QGSA and the area along Dahepiaoliu. Both are tourist areas with limited space, abundant tourist resources, a high volume of tourists, and numerous tourist facilities.

*4.3. Risk Assessment of Geological Disasters*

Building upon the geological disaster hazard assessment and vulnerability assessment described earlier and aligning with the definition of geological disasters risk degree as the product of hazard degree and vulnerability, this paper, utilizing the GIS platform, recalibrates values for the hazard grid and vulnerability grid. Subsequently, it computes the risk assessment grid of QGSA through raster algebra multiplication. Employing the natural break method within the GIS platform, the product is categorized into four risk value intervals: 1~2, 3~4, 5~9, and 10~16. Higher risk values signify a greater risk of geological disasters in the area. By merging the risk value intervals with the actual conditions of QGSA, cluster analysis is applied to segment the area into four regions: extremely high-risk, high-risk, medium-risk, and low-risk areas (Figure 13a). The proportions of each zone are statistically determined (Figure 13b,c), revealing that the low-risk area spans 5.45 km$^2$, constituting 53.66% of the entire scenic area; the medium-risk area covers 3.59 km$^2$, accounting for 35.43% of the total area; the high-risk area encompasses 0.72 km$^2$, representing 7.14% of

the overall area; the extremely high-risk area extends over 0.38 km$^2$, making up 3.77% of the entire scenic area.

**Figure 13.** Risk assessment zoning result in QGSA. (**a**) Risk assessment zoning; (**b**) area chart; (**c**) area percentage chart.

Among them, the extremely high-risk area is mainly distributed around the main scenic area of Qinglong Gorge, presenting a strip-like spatial distribution (Figure 13a). This area is rich in tourist resources, with a tourist resource density of up to 29 sites/km$^2$. It is the most densely populated area with tourists, scenic area staff, and various tourism facilities. There are a total of 11 geological disasters developed in this area, with landslides being the predominant hazard. These hazards pose a certain threat to the personal and property safety of tourists and scenic area staff. In an environment with a high disaster risk, a relatively large population exposed to hazards, and intensive resources, the geological disaster risk in this area is extremely high.

The high-risk area mainly includes small villages along provincial road S327 and within the Qinglong Gorge tributary (Figure 13a). This area has diverse landforms, including erosional low mountain areas, erosional middle mountain areas, and alluvial river valley areas. There are numerous residential buildings and roads, with a tourist resource density of up to 17 sites/km$^2$. The economic attributes of the disaster-affected population in this area are prominent, and the population is relatively dense, making it a high-risk area for geological disasters.

The medium-risk area is mainly distributed in small villages such as Dongzhuangjie, Xijie Shang, and Shuiquanshang on both sides of the valley (Figure 13a). The landforms in this area are mainly erosional low mountain areas, with a small amount of alluvial river valley areas and erosional middle mountain areas. There are no large-scale residential areas in this region, with only a few scattered residential buildings on both sides of the provincial road. There is only one tourist area, Nuwa Cave, which has been abandoned and not open to tourists for a long time. Therefore, this area is a medium-risk zone for geological disasters.

The low-risk area is mainly distributed in the periphery of QGSA (Figure 13a), including several very small villages such as Qianqian Village, Lingdong Village, and Dalinbei. The landform is mainly erosional middle mountain areas, and the vast majority of the area is uninhabited. There are only three tourist resources in this area, and there are very few residential buildings and tourist facilities. The economic attributes of the disaster-bearing body are weak. Therefore, this area is a low-risk zone for geological disasters.

*4.4. Countermeasures of Geological Disaster Prevention and Control in QGSA*

QGSA primarily features deep-cut gorges and unique rocky landscapes. While addressing disaster management, it is essential to emphasize the preservation of the original beauty of the landscape. Therefore, effective engineering control and risk management measures need to be implemented in QGSA. Building on the geological disaster risk assessment, this paper proposes a risk classification and control system that has been proposed for prevention and control. The areas of extremely high to high risk are mostly located in the core areas of QGSA and along the S327 provincial road, characterized by high population density, extensive tourism development, and frequent human engineering construction. Therefore, this region is designated as a key area for governance and control, implementing geological hazard engineering control measures that combine passive and active protection. Areas with moderate to low risk have fewer human activities and moderate levels of tourism development. However, they are closer to residential areas, tourist attractions, and roads and bridges. For larger-scale geological disasters that affect buildings and human activities, specific control measures should be taken. Furthermore, efforts should be made to disseminate knowledge about geological disasters and raise awareness among residents about geological disaster prevention within the area.

## 5. Conclusions

(1) Exploring the calculation methods of geological disaster risk elements, the applicability of these calculation methods is clarified.

During the assessment of indicators for geological disaster hazard and vulnerability in QGSA, we conducted a comprehensive exploration of various methods for calculating elements. Specifically, we employed empirical, statistical, and theoretical approaches to determine the weights of geological disaster impact factors. This investigation aimed to provide a clear understanding of the suitability and applicability of each method in the assessment of geological disaster risk.

(2) By using the GIS platform, geological hazard risks and vulnerability assessment indicators for QGSA are proposed.

Research on QGSA encompasses the use of regression models and the Analytic Hierarchy Process (AHP) to establish risk indicators for geological disaster hazard and vulnerability, providing the foundation for risk assessment. We selected three categories of factors, including historical disasters, formation conditions, and triggering conditions, totaling seven factors, to create a logistic model for assessing geological disaster hazards. We determined the corresponding indicator coefficients to calculate hazards in QGSA. Similarly, we utilized three categories of factors, including population vulnerability, material vulnerability, and vulnerability of tourist resources, with a total of seven factors, in the AHP method to construct a vulnerability assessment model for geological disasters. The corresponding indicator coefficients were determined to calculate the risk in QGSA. Subsequently, we assessed the risk in the scenic area according to the geological disaster risk theory, categorizing the area into low-risk, medium-risk, high-risk, and extremely high-risk zones. The low-risk area covers 5.45 km$^2$, accounting for 53.66% of the total area; the medium-risk area covers 3.59 km$^2$, representing 35.43% of the entire scenic area; the high-risk area spans 0.72 km$^2$, accounting for 7.14% of the total area; and the extremely high-risk area encompasses 0.38 km$^2$, making up 3.77% of the entire scenic area. The applicability of these research methods to risk assessment in the scenic area has been validated through field investigations.

(3) In light of the risk assessment results, a prevention and control system is proposed for high-risk areas, including monitoring and early warning, and engineering control of potential hazards.

Due to the long-term effect of the topographic conditions of the canyon scenic area, according to the risk assessment results, the extremely high-risk and high-risk area is along

the Yixiantian to Qingren Lake, Dahe Piaoliu, Dahe Village, and Hedong Village. The dangerous rock mass and loose solid debris material generated by rock mass cracks are easily transformed into new geological disasters to form a disaster chain under heavy rainfall conditions. In order to ensure the safety of tourists, it is recommended to carry out effective project governance, monitoring, and risk control of this key area.

(4)  The findings from this research paper suggest that numerical simulation methods can be employed for geological disaster risk assessments both before and after implementing the proposed prevention and control measures to validate their feasibility and effectiveness.

**Author Contributions:** Conceptualization, R.M. and Y.L.; Software, R.M.; Resources, T.C.; Data curation, Q.Z.; Writing—original draft, R.M. All authors have read and agreed to the published version of the manuscript.

**Funding:** Investigation and research of geological landscape and high-quality construction in the Yellow River Basin (300102262908); Experimental study of granite earthquake landslide shaking table (SKLGP2018K015); Study on the mechanism of granite landslide in the middle of the northern margin of the Qinling Mountains (41672285).

**Institutional Review Board Statement:** Not applicable.

**Informed Consent Statement:** Not applicable.

**Data Availability Statement:** Data are contained within the article.

**Conflicts of Interest:** The authors declare no conflict of interest.

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
