# Peer review of "Preliminary Risk Assessment of Geological Disasters in Qinglong Gorge Scenic Area of Taihang Mountain with GIS Based on Analytic Hierarchy Process and Logistic Regression Model"

_sustainability, doi:10.3390/su152215752_

Round 1
Reviewer 1 Report
Comments and Suggestions for Authors
This paper develops a GIS-based geological risk assessment based on logistic regression and analytic hierarchy process for the Qinglong Gorge Scenic Area (QGSA). The topic is relevant and of international interest as it addresses the complex relationships between natural hazards and human activities that influence the behaviours of natural phenomena and contribute to increase risk levels due to the vulnerability aspects.
However, the structure of the paper can be improved as well as some contents. First of all, the scientific relevance of the topic could be better explained by citing more recent international studies and international documents such as the UN Sendai Framework for Disaster Risk Reduction 2015-2030. I suggest examining the statement "Due to the uncertainty of geological disasters occurrences and their consequences (...)" (l. 1-52) before the literature review focused on methods adopted and test cases (l.53-85).
Also, a methodological section could be added at the beginning of the section "2. Development characteristics and influencing factors of geological disasters" (or before section 2) to explain how the disasters were identified, the influencing factors were defined and they were combined and the vulnerability factors were selected and weighted.
Then, there are some specific comments below:
L. 27-31 This part of the abstract does not reflect the results of the paper which focuses on risk analysis to support monitoring and early warning systems rather than the proposal of a monitoring and early warning system. This also needs to be specified in section 3.4.
Also, “comprehensive monitoring and early warning, risk classification control, engineering management and landscape coordination” seems to be quoted but there is no reference. See also L.431-433
L. 35-91: This can be considered the introduction but it directly introduces the case study in the first part (further addressed in section 1) and the references for risk assessment methods in the second part. This structure can be revised by combining the aspects referred to the overview of the case study in section 1 and the theoretical-methodological background.
L. 39 Please provide a footnote to explain what “4A tourist rating” means.
l. 75 Is “Geographic Information Systems (GIS), first proposed by Garrison (…)” a reference?
L. 120 – Figure 1: It is not clear whether the information reported (landslides, rockfalls) in this map is the result of the investigations described in section 2 (L. 123-125) or obtained from other studies or databases. It may be moved below and specified.
Section 2 appears to directly show the results of the analysis. It should be explained how the influencing factors were chosen and whether there are references to support the selection process.
L. 227-228: I suggest changing the title: hazard has been clearly defined (L. 55-61) as a component of risk while danger is never mentioned. I suggest carefully checking the use of the terms hazard, risk, evaluation or assessment, etc. by looking at the UNDRR Terminology (https://www.undrr.org/terminology)
L. 263: Does this section present the results of the hazard or risk analysis? The title is confusing as the final section 3.3. actually presents "Risk evaluation of geological disasters".
Figure 8 is correctly titled "Geohazard assessment zoning result in QGSA" but then in the legend there is again the term danger.
L. 346-352: How are these factors and factors weights determined? Have other similar studies been considered?
L. 442 The section presents the main outcomes of the work in a clear and concise way, however it is also appropriate to underline the steps forward in knowledge gaps filled by this research (lack of data, or lack of analysis, etc.), limitations and validation of the results, future developments and replicability of the analysis (common databases, suitability for a local scale or multi-scalar method).
Comments on the Quality of English LanguageThe manuscript needs proof-reading before re-submission.
Author Response
Refer to the attachment

Reviewer 2 Report
Comments and Suggestions for Authors
The authors clearly defined the research problem and established models for assessing geological risk as a final result of their survey. As a starting point, they included 52 collapse disasters and 17 landslides.
As a base for the methodological research, they use a GIS Base, and Analytic Hierarchy Process, and a Logistic Regression Model.
The literature review is based on 38 references, academic articles, and relevant public data.
All around research and developed models can be deeply researched and applied in any region as a good base to prevent disasters or minimize damages.
Comments on the Quality of English LanguageJust a proofreading, all around everything is ok.
Author Response
Refer to the attachment

Reviewer 3 Report
Comments and Suggestions for Authors
Dear Authors,
"The paper titled 'Geological Risk Assessment and Vulnerability Analysis in the Taihang Mountain Area Using GIS and Statistical Models' delves into a topic of significant relevance, examining the intricate relationship between geological hazards, risk assessment, and vulnerability analysis, specifically in the unique context of the Taihang Mountain region."
The article explores geological risks and vulnerability in the Taihang Mountain area using a Geographic Information System (GIS) and statistical models. The authors focus on identifying geological threats, creating a risk model, and assessing the vulnerability of this region. The research design, questions, and hypotheses are precisely defined. The authors have also developed a risk model and a vulnerability model, adding value to their work.
"In particular, the article excels in several areas. Firstly, it contains relevant research information and effectively employs various methods and analyses to substantiate its conclusions. This meticulous approach adds a layer of credibility to the findings, reinforcing their significance.
Secondly, the article offers a comprehensive analysis of geological hazards in the specified area, harnessing the power of Geographic Information Systems (GIS) to assess these hazards. This utilization of GIS not only enhances the precision of the assessment but also highlights the authors' commitment to employing contemporary tools for rigorous research.
Furthermore, the inclusion of a diverse array of factors for assessing the risk of geological hazards is commendable. The incorporation of historical data, formation conditions, and triggering conditions into the risk assessment model enriches the depth of the study, ensuring a more holistic understanding of the landscape's vulnerability to geological disasters.
Lastly, the article stands out in its presentation of the results derived from the risk and vulnerability assessment of geological hazards in the area. The categorization into different zones based on the level of risk adds a practical dimension to the study, making it potentially instrumental for policymakers, disaster management authorities, and other stakeholders."
However, there are areas where the article could be improved.
1. In some cases, more details about the methodology used for risk and vulnerability assessment could be beneficial.
2. It could be useful for the article to include more discussion on how these results could be applied in practice or their significance for the region.
3. Suggestions for future research are a crucial part of the conclusion of a scientific article. It indicates what questions or topics could be further explored based on the findings in the article. These recommendations for future research can assist other researchers in deciding where to direct their research efforts.
It's advisable for the authors to consider possibilities for future research related to the study's topic in the concluding section, even though it may not be mandatory.
Overall, the article has potential and provides valuable information about geological risks in the Taihang Mountain area.
Good luck in your future research
Author Response
Refer to the attachment

Reviewer 4 Report
Comments and Suggestions for Authors
Risk Assessment of Geological Disasters in Qinglong Gorge Scenic Area of Taihang Mountain with GIS Based on Analytic Hierarchy Process and Logistic Regression Model by Ma et al. This article has strong logic, rich data used for simulation, and mature methods used. The following are the review comments for:
1. The citation of the paper is very non-standard, such as lines 61 and 71. In addition, the article lacks necessary references, such as in the first paragraph. In summary, the author is requested to make revisions according to the publisher's requirements.
2. Line 19. "evaluate the geohazard of geological disasters", please confirm that this sentence is correct.
3. Please add coordinate information in Figure 1.
4. Please remove the Chinese characters in the images.
5. Missing vertical axis in Figure 8-b.
6. The introduction of the article is relatively ordinary. It only lists the research results of predecessors and lacks summary and induction. The research content was not well introduced in this study.
Author Response
Refer to the attachment

Reviewer 5 Report
Comments and Suggestions for Authors
The manuscript on geological disaster risk and vulnerability assessment in the Qinglong Gorge Scenic Area (QGSA) holds promise but requires enhancements to strengthen its contribution to regional disaster management and resilience. Here are key areas for improvement:
· Geological and Environmental Context: The manuscript needs to include geological data and discuss the local factors that render this area susceptible to hazards. Delve into how climate change, with its evolving weather patterns, precipitation levels, and temperature fluctuations, might influence geological risks over the long term.
· Field Survey Details: Enhance the clarity of field survey processes by detailing the timing, methods employed, and a direct comparison with an existing geohazard inventory. To augment the practical value of the paper, incorporate historical disaster records and photographs of the hazards observed during the study field survey. Ensuring model calibration with historical data specific to QGSA will further strengthen the research's credibility.
· Advanced Remote Sensing Techniques: Mentioned but not yet elaborated, using advanced remote sensing techniques and technologies like high-resolution satellite imagery and LiDAR data can significantly enhance hazard detection accuracy. Articulate how these modern tools can improve the identification and monitoring of geological hazards.
· Uncertainties and Limitations: A transparent discussion of the Regression model's inherent uncertainties and limitations is essential. This transparency ensures a comprehensive understanding of the research's validity, which is critical for its acceptance and applicability.
· Methodological Schema: Consider adding a methodological schema in a figure format to represent the research process visually. This visual aid will significantly enhance readers' comprehension of the study's framework.
· Scale Considerations: While the paper mentions large-scale field surveys and mapping, ensure that the evaluation factors align with the local scale of the QGSA region. Analyzing a local area with large-scale maps may not be appropriate, and addressing this consideration will strengthen the applied methodology.
· Scenario Analyses: Given the evolving nature of vulnerabilities over time, exploring scenario analyses to assess the potential impacts of geological disasters at different return periods is relevant. The involvement of domain experts and stakeholders in scenario development and validation is vital to enhancing the model's reliability and real-world relevance.
By addressing these points, the manuscript can contribute to disaster management practices in the study area, benefiting researchers and practitioners alike.
Author Response
Refer to the attachment

Round 2
Reviewer 1 Report
Comments and Suggestions for Authors
The Authors addressed many of the weak points of the paper, however they did not fully consider the general comments regarding the need to revise the structure of the manuscript, by clearly explaining the methodology and detailing the first section. Below are my additional suggestions.
First, the paper begins with a description of the area, which is then further described in section 1, and jumps directly to reviewing the methodologies. A theoretical framework of the topic is missing. Why is disaster risk assessment scientifically relevant? What do the international documents say? The key to understanding disaster risk is in the sentence "Due to the uncertainty of geological disasters occurrences and their consequences (…)" (l. 52-54). As a result, it is appropriate to dedicate the first section to the theoretical framework and the review of the methodologies and then introduce the contents of the following sections: the choice of method compared to others, the case study, and the expected results.
Second, it is not yet clear whether all the starting information was constructed from scratch or deduced from existing databases and studies, nor what progress or limitations have influenced the methodology and its application. An example, where does the information in figure 1 on geological structure and geological disaster distribution come from? For this reason, I continue to suggest the addition of a section for the presentation of the methodology which can include aspects of the starting situation, also to give greater relevance to the achieved results. The addition to lines 132-138 is not enough to explain all the steps. Also, if there are previous studies in the same area that have provided preparatory information for this study, in terms of methods or results, they must also be cited to give continuity to the research.
Lastly, after the improvement of the previous sections, section 4 (Conclusion) may be strengthened.
Comments on the Quality of English LanguageMinor editing of English language required after the review
Author Response
Dear Reviewer:
We sincerely thank the editor and all reviewers for their valuable feedback that we have used to improve the quality of our manuscript. See the attachment for detailed response.

Reviewer 5 Report
Comments and Suggestions for Authors
The authors' efforts to address the previous review comments of the manuscript, titled "Risk Assessment of Geological Disasters in Qinglong Gorge 2 Scenic Area of Taihang Mountain with GIS Based on Analytic Hierarchy Process and Logistic Regression Model" are acknowledged. However, significant issues still require attention, along with the need for revisions to the justifications provided.
First, the paper requires incorporating the geographical scale and specify the return period for risk analysis. These are crucial pieces of information since risk evolves with changing vulnerability conditions. Defining the scale is particularly important to help readers contextualize the study's significance.
Second, a comprehensive review of references is essential. Some references cannot be located, and the citation format needs to be more consistent. Additionally, the disproportionately high number of references on a single page out of 17 disrupts the paper's overall structure.
Third, the authors should address the issue of overly general conclusions that lack specific support, rendering them less impactful. Conclusions should be robustly underpinned by evidence.
Lastly, the authors must reconsider the current presentation of Figure 2 and Figure 6. While the authors justify their relevance, these figures do not adequately represent the fieldwork conducted. It is recommended to resize those figures and include their location in the hazard map and add additional photos or evidence of the validation in the field. Additionally, if fieldwork evidence is inconclusive, the title must reflect this including "Preliminary Risk Assessment"
For these reasons, a major revision is strongly recommended to align the paper with the publication standards expected.
Author Response

(The authors gave the same response as above.)

Round 3
Reviewer 1 Report
Comments and Suggestions for Authors
The manuscript has been significantly revised. It is now clear how the preliminary analysis was carried out through the addition of section 1.2. However, the introduction can still be improved. Since the addition of a methodological section was not considered necessary by the Authors, it is even more important that this is clearly anticipated in the introduction. I suggest you refer to other similar scientific articles and address the following points:
- General theoretical introduction. What is the topic, why it is relevant in scientific debate, and what are the existing gaps to fill?
- Background: literature review
- Introduction of the case study and method
Also, the conclusion section is still weak. Using bullet points helps to fix the main outcomes, but it is always suggested to support it with at least one introductory sentence and a final comment.
Because of the double round of revisions, the paper needs a general proofreading. In particular, lines 51-62 must be better connected to the next part, avoiding repetitions and giving continuity to the reading.
Author Response
Specific answers can be found in the attachment.

Reviewer 5 Report
Comments and Suggestions for Authors
The authors' efforts in addressing the previous feedback are recognised. However, the manuscript still requires significant improvements to meet the journal's expected standards.
Concerning the figures, several issues have been noted. Figures 1, 2, and 3 need to include crucial details and an accurate representation of fieldwork. Figure 1 is not a geological or structural geology map. Figure 2: Captions and footnotes should clarify the methods used in data collection and analysis. The texts in Figure 3 are not readable, and the caption is wrong (they are not 'geological disasters'). Further, Figures 4 to 6 need unit labels for clarity, and Figure 8 demands a review of the density map and an explanation of the selection of ranges for accuracy.
Furthermore, the methods section needs more precise explanations, particularly concerning regression coefficients, risk assessment, and data precision. Specifying the return period for risk analysis is crucial, as it influences the evolving risk dynamics. Additionally, due to the uncertainties detected, including a section on method limitations and data collection is essential for transparency and credibility, as is changing the title to include "preliminary assessment".
The abstract requires revision for better coherence with the title and should highlight the key research points.
For further reference on similar research, please consult the provided examples:
Ali, S. A., Mohajane, M., Parvin, F., Varasano, A., Hitouri, S., Łupikasza, E., & Pham, Q. B. (2023). Mass movement susceptibility prediction and infrastructural risk assessment (IRA) using GIS-based Meta classification algorithms. Applied Soft Computing, 145, 110591.
Martha, T. R., van Westen, C. J., Kerle, N., Jetten, V., & Kumar, K. V. (2013). Landslide hazard and risk assessment using semi-automatically created landslide inventories. Geomorphology, 184, 139-150.
Comments on the Quality of English Language
Language usage in the manuscript requires thorough scrutiny by native English speakers. For instance, the excessive use of the word "based" is noticeable, appearing 39 times throughout the text. Considering the limited text content of approximately 12 pages, this term is reiterated roughly four times on each page.
Author Response

(The authors gave the same response as above.)
